# Metformin and Probiotics in the Crosstalk between Colitis-Associated Colorectal Cancer and Diabetes in Mice

**DOI:** 10.3390/cancers12071857

**Published:** 2020-07-10

**Authors:** Sahar Al Kattar, Rosalyn Jurjus, Aline Pinon, David Yannick Leger, Abdo Jurjus, Chawki Boukarim, Mona Diab-Assaf, Bertrand Liagre

**Affiliations:** 1Laboratoire PEIRENE EA 7500, Faculté de Pharmacie, Université de Limoges, 2 Rue du Docteur Raymond Marcland, 87025 Limoges Cedex, France; sahar_kattar@hotmail.com (S.A.K.); aline.pinon@unilim.fr (A.P.); david.leger@unilim.fr (D.Y.L.); 2Doctoral School of Sciences and Technology, Lebanese University, Hadath 99000, Lebanon; mdiabassaf@ul.edu.lb; 3Department of Anatomy, Cell Biology and Physiology, American University of Beirut, Beirut 1107-2020, Lebanon; rj58@aub.edu.lb; 4Department of Chemistry, Faculty of Sciences III, Lebanese University, Mont Michel Ras Maska, El Koura 826, Lebanon; cboukarim@ul.edu.lb

**Keywords:** colorectal cancer, microbiota, inflammation, diabetes, oxidative stress, probiotics

## Abstract

The co-occurrence of colorectal cancer (CRC) and diabetes mellitus along with inflammation and dismicrobism has been frequently reported. Several studies shed light on the antioncogenic potential of metformin on colorectal carcinogenesis. This study aimed to demonstrate that metformin in association with probiotics acts in a synergic effect in breaking the crosstalk, thus inhibiting CRC progression, improving diabetes, and reducing inflammation. Ninety-six male Balb/c mice, 6–8 weeks old, were divided into 16 control and experimental groups to assess the effect of the different treatments and combinations at the clinical, histological, and molecular levels. Metformin and probiotics showed beneficial outcomes on CRC and diabetes, alone and most importantly in combination. Their effects were exerted by inhibiting the inflammatory process whereby a downregulation of IL-6 and TNF-α as well as oxidative stress were depicted. The characterization of the effects of probiotics and metformin on CRC and diabetes sheds light on the role of inflammation and microbiota in this crosstalk. Deciphering the downstream signaling pathways elicited by these compounds will help in developing new effective targeted treatment modalities.

## 1. Introduction

Colorectal cancer (CRC) is the third most commonly diagnosed cancer worldwide with 1.8 million new cases and almost 881,000 deaths in 2018, according to the World Health Organization GLOBOCAN database [1]. In Lebanon, CRC accounts for 8.5% of all cancers. It is one of the highest incidence rate in the MENA region, as 1109 cases of CRC were diagnosed in 2018 in a population of almost 6 million [1,2].

It is well established that the etiology of CRC is multifactorial, encompassing genetic, environmental, and lifestyle-related factors, including westernized diet, alcohol consumption, and obesity among others. However, chronic inflammation, in particular inflammatory bowel disease (IBD) and dysbiosis in enteric microbiota, remain the key players in this process [3,4,5,6]. Additionally, a consensus statement by the American Diabetes Association and the American Cancer Society in 2010 concluded that there is a higher risk of CRC among patients with diabetes mellitus, shedding more light on diabetes as a risk factor for CRC [7].

Diabetes mellitus is a group of metabolic disorders characterized by chronic hyperglycemia and dysregulated metabolism of carbohydrates, lipids, and proteins induced by insulin insufficiency [8]. According to the International Diabetes Federation, diabetes affected 451 million people worldwide in 2017; this number is expected to rise to 693 million by 2045. This rising trend has been promoted by a shift into urban lifestyle, the spread of western style diet, and lack of physical activity [9].

A role of microbiota in the development of diabetes mellitus has been recognized; several studies showed that in diabetic humans, there is a lack of uniformity in gut microbiota profiles, as these individuals show differences in both human host markers and gut microbiome signatures when compared with healthy people [10,11]. It has also been suggested that there might be an inflammation-triggering effect of the intestinal microbiota in the development of autoimmune diabetes [11,12]. Previous publications from our group and others supported the possible pathogenic links among CRC and diabetes, whereby an altered glucose metabolism, an oxidative stress, and a chronic inflammatory state, triggered by an imbalance in the intestinal microbiota, are potential mediators [3,13,14]. 

It has been also reported that inflammation is a common denominator between CRC and diabetes mellitus and a crucial factor contributing to the development of these disease entities, as subclinical systemic inflammation has been observed in patients with diabetes and in patients with CRC. It is characterized by elevated circulating levels of inflammatory parameters, including C-reactive protein and inflammatory cytokines such as interleukin-6 (IL-6) and tumor necrosis factor-Alpha (TNF-α), among others [15]. IL-6 and TNF-α, are the main MAPK-activated protein kinase 2 mediated cytokines. They are strongly associated with inflammation-driven cancers, including CRC and contribute to tumor growth and invasion [16]. Moreover, these inflammatory disease entities are characterized by excessive reactive oxygen species (ROS) production that triggers inflammation by driving the production of proinflammatory cytokines [3,17]. 

On the other hand, the human gut microbiota is a large and complex microbial community, comprising different populations of microorganisms that live in the intestinal lumen: bacteria, viruses, fungi, archaea, bacteriophages, and protozoans, with bacteria being the most abundant. Approximately 100 trillion bacteria inhabit the human intestine, a count ten times greater than the total number of cells in human body, with at least 1000 different species of known bacteria whose genome is 150 times greater than that of humans [18,19]. Microbial distribution varies spatially and temporally from the mouth to the rectum during the individual’s lifetime, and several host factors are able to shape the microbial community, including genetics, the mode of delivery at birth, and antibiotics use, among others [19]. 

However, a perturbation of the normal gut microbial balance, defined as dysbiosis, can disrupt the intestinal barrier and its associated immune function, thus leading to autoimmunity and chronic inflammation [5,20]. This chronic inflammatory state promotes cell proliferation, angiogenesis, and apoptosis, along with the production of an array of proinflammatory cytokines, growth factors, and reactive oxygen and nitrogen species that result in the generation of a carcinogenic microenvironment; in addition to extra intestinal manifestations, affecting the host metabolism and promoting obesity, insulin resistance, and diabetes [5,21,22,23].

A series of recent publications support the fact that inflammation paralleled with dysfunctional interactions between gut microbiota and the mucosal immune system are well defined risk factors for IBD, CRC, and other inflammatory and metabolic diseases such as diabetes [3,5,13]. On this basis, the recommendation of probiotics as part of the management protocols has been widely considered [23,24]. 

The World Health Organization defines probiotics as live microorganisms that when administered in appropriate amounts, confer a health benefit on the host [25]. In the last few years, a growing interest in studying and using probiotics has been observed for the treatment of gastrointestinal diseases and the improvement of overall human health [26]. The leading mechanism of probiotics action primarily includes alteration in the composition of gut microbiota, maintaining epithelial barrier function, competition with harmful gut flora for nutrients and adhesion to the epithelium of the gastrointestinal tract, as well as the production of antimicrobial peptides that were proven to exhibit anticarcinogenic action, anti-inflammatory effects and anticholesterol activities [4,25,27].

On the other hand, metformin, a widely prescribed antihyperglycemic agent of the biguanide family, is one of the most extensively recognized metabolic modulators with well-documented anticancer properties [28,29]. Metformin has a high oral bioavailability and its mechanisms in targeting the host for the treatment of metabolic diseases have been well studied. However, recent studies revealed that the gut microbiota might play an important role in its efficacy, shedding light on the fact that the gastrointestinal tract is a key site in the action of metformin [30,31]. Thus, the modulation of the gut microbiota may be one of the mechanisms contributing to the antidiabetic and anticarcinogenic effects of metformin.

The challenge to treat cancer still lies in the inability to identify new treatment strategies beyond chemotherapy, immunotherapy, and surgery to inhibit tumor progression. Given the importance of the gut microbiota and dysbiosis in the crosstalk between CRC and diabetes, our present study evaluated the effect of the addition of a microbiota modulator on metformin in a well-established model of colitis-associated CRC that recapitulates the progression from chronic inflammation to dysplasia and adenocarcinoma in humans, with an emphasis on the role of diabetes in this process. This work explored the molecular mechanisms of metformin and probiotics while focusing on their chemopreventive properties and the importance of microbiota in this strategy.

## 2. Results

### 2.1. Effect of Metformin and Probiotics on Glycemia

All the data, clinical, histological, immunohistochemical, and molecular, were analyzed with a focus on the effect of the combination therapy of metformin (M) and probiotics (P) in treating CRC in nondiabetic and diabetic animals. Diabetes was successfully induced in all intended animals. Blood glucose level was measured weekly as a direct indicator of diabetic status.

As expected, glycemia levels had increased beyond 250 mg/dL one week after an IP injection of 150 mg/kg Streptozotocin (STZ). These high levels were successfully maintained in all diabetic untreated mice (group D and DCRC) throughout the experimental period (Figure 1b).

All animals belonging to the groups where diabetes was not induced, i.e., in groups NC, CRC, CRC + M, CRC + P, CRC + MP, M, P, and MP, had glycemia levels below 250 mg/dL at all time points and the difference between these groups was not statistically significant (ns) as shown in Figure 1a. However, in the CRC group, whose animals were not subjected to diabetes induction by STZ, high glycemia peaks were observed in weeks 3, 5, 8, and 11 in which Dextran sulfate sodium (DSS) was administered. These blood sugar fluctuations were only significant at week 8 when comparing normal controls (NC) to the CRC group. The metformin and probiotics (MP) combination significantly reduced the peaks at week 8 (CRC + MP versus CRC, ^#^
*p* < 0.05) as shown in Figure 1a.

The highest glycemia values were obtained in DCRC and D groups at all time points with no statistical difference (ns) between the two groups showing that CRC did not affect glycemia levels in these animals. In diabetic animals, P and M administration alone or in combination significantly reduced glucose levels during the experimental period. In comparison with the untreated diabetic mice, with CRC induction (DCRC group), M and P single drug treatment in DCRC + M and DCRC + P groups decreased significantly blood glucose levels (Figure 1b, * *p* < 0.05). Interestingly, M and P combination showed a significant effect, greater than either drug alone, as depicted when comparing DCRC + M versus DCRC + MP groups (^†^
*p* < 0.05) and DCRC + P versus DCRC + MP groups (^†^
*p* < 0.05), indicating that P helped M in alleviating the hyperglycemic phenotype induced by STZ (Figure 1b). 

### 2.2. Metformin and Probiotics Modulate CRC Induction in Balb/c Mice

The optimal procedure adopted for CRC induction was successful in all mice. DSS concentration and its number of cycles were determined by pilot studies using different concentrations of DSS ranging from 1% to 3%, whereby treatment of male Balb/c mice with 1.5% DSS in their drinking water for four cycles, in addition to an injection of 10 mg/kg Azoxymethane (AOM), resulted in clinical signs and symptoms and gross and histological alterations associated with CRC. 

Animals that were subjected to this optimized DSS/AOM protocol showed signs of sickness starting the first cycle, and these signs were aggravated by each successive DSS cycle. The main observed alterations were weight loss (Figure 2), loose stools, diarrhea, and rectal bleeding (Table 1), as calculated in the disease activity index (DAI). These animals presented signs of discomfort and sickness such as bad posture, hunched back, decreased grooming, and low mobility and responsiveness.

Nondiabetic and diabetic animals that were not subject to CRC induction had an increase in their average weight as seen in the NC, M, P, MP, D, D + M, D + P, and D + MP groups. On the other hand, weight loss was detected in groups with CRC; the lowest weight averages were recorded in CRC and DCRC groups, with no statistically significant difference between these two groups. Treatment with metformin or probiotics alone and, most importantly, in combination, had a positive effect in preventing the weight loss in the nondiabetic and diabetic CRC animals with **^#^**
*p* < 0.05 and * *p* < 0.05 respectively (Figure 2).

When looking at fecal occult blood, normal controls and groups that were not subjected to CRC induction (NC, D, M, D + M, P, D + P, MP and D + MP groups) showed negative occult blood for all animals at all time points. Groups that were subjected to CRC induction showed positive results for fecal blood to varied degrees. CRC and DCRC animals were the first to show blood in their stools in 17% of the animals starting week 3 (on the first DSS administration), to reach 100% by the end of the experiment (Table 1).

In line with the improvements encountered in the glycemia levels and body weight, treatment with P and M alone or in combination reduced the frequency of blood in the stools and delayed their appearance until week 5. The best scores were obtained with the combination treatment in nondiabetic and diabetic CRC animals (CRC + MP and DCRC + MP groups) whereby appearance of blood was observed in only 67% of the animals at week 5 and the positive rates were 50% and 33%, respectively, at week 13 (Table 1). Moreover, when comparing single drug treatment in nondiabetic and diabetic CRC animal groups (CRC + M, CRC + P, DCRC + M and DCRC + P) to the combination groups (CRC + MP and DCRC + MP), lower percentages in the combination treatment were encountered (Table 1). Treatments triggered a decrease of the blood in the stools, thus indicating a recovery in the mucosa.

In the CRC and DCRC groups, animals had the highest disease activity indexes (Figure 3). Treatment of CRC animals with M alone or P alone ameliorated the clinical profile of the animals and decreased DAI, but not significantly. Importantly, the combined MP treatment induced a statistically significant decrease in DAI levels in nondiabetic CRC and diabetic CRC (^#^
*p* < 0.05 and * *p* < 0.05, respectively), as these groups (CRC + MP and DCRC + MP) had the lowest DAI along with a better mobility, fur shape, and responsiveness. The addition of P to M induced a significant amelioration in its action on decreasing DAI in nondiabetics (CRC + M versus CRC + MP, ^†^
*p* < 0.05) and in diabetics (DCRC + M versus DCRC + MP, ^†^
*p* < 0.05) at week 13 (Figure 3).

Normal DAI levels close to zero were obtained in normal controls (NC group) and in all of the other groups that were not subject to AOM/DSS CRC induction (M, P, MP, D, D + M, D + P and D + MP groups). Treatment of normal mice with P alone, M alone, and their combination did not affect the animals, their DAI was similar to those of normal mice at all time points. Diabetic animals showed more signs of discomfort and sickness than their nondiabetic counterparts; however, their diabetic status did not seem to affect their DAI score and no significant differences between diabetics and nondiabetics were observed (Figure 3).

### 2.3. Effect of Metformin and Probiotics on the Survival Rates

Based on our experience and the reported literature, week 13 was selected as the terminal time point. Survival rates fluctuated between the different groups based on treatments. In diabetics, the survival rate of diabetic animals with CRC induction was the lowest at 50% (DCRC group). Treatment with M and P improved survival rates to 67% and 83% in the DCRC + M and DCRC + P groups, respectively. Moreover, the MP combination in the DCRC + MP group improved and raised the survival rate to 100%.

Nondiabetic animals that were subjected to CRC induction had survival rates of 67%; while the same animals treated with M and P alone or in combination (CRC + M, CRC + P, and CRC + MP groups) had 100% survival rates, similar to normal animals in the normal controls (NC) and in nondiabetic M, P, and MP groups (Figure 4).

### 2.4. Metformin and Probiotics Restore Normal Colon Length

Colon length was measured upon dissection at week 13 as colon shortening is often considered a visual index that reflects the severity of colorectal inflammation. Mice treated with AOM/DSS (CRC and DCRC groups) had the shortest colons with 6.75 ± 0.78 cm and 6.67 ± 0.17 cm, respectively, (Figure 5). 

In nondiabetic animals that were subject to CRC induction (CRC group), mice had significantly shorter colons than the normal controls (NC) (^#^
*p* < 0.05). Administration of M in the CRC+M group increased the colon length, but this was not statistically significant. However, administration of P alone in (CRC + P) group or in combination with metformin (CRC + MP) significantly increased colon length to reach 10.58 ± 0.83 cm and 10.83 ± 0.31 cm, respectively, with ^#^
*p* < 0.05 (Figure 5), close to that of the normal control (NC) group. 

Diabetic mice with CRC (DCRC group) had significantly shorter colons than diabetic animals (D group), * *p* < 0.05. The administration of M or P alone ameliorated the colon length, but not significantly in the DCRC + M and DCRC + P groups. However, administration of MP in combination restored the normal length of the colon, whereby animals in DCRC + MP group had colon length of 11 ± 0.63 cm (* *p* < 0.05) (Figure 5). 

Colon length was not affected by diabetes alone, as the difference between nondiabetic and diabetic animals in all subgroups was not statistically significant.

### 2.5. Effect of Metformin and Probiotics on Polyp Formation

Polyp formation was assessed by quantifying the number of formed nodules within the colon from the ileocecal junction to the distal end of rectum. 

Animals in the CRC and DCRC groups had an average of 9.25 ± 0.85 and 10.33 ± 1.76 polyps, respectively. This number was significantly higher than the negligible number obtained in the normal controls (0.67 ± 0.33), with ^#^
*p* < 0.05 and * *p* < 0.05, respectively.

Treatment with either drug alone in nondiabetic and diabetic CRC animals reduced polyp formation; however, this reduction was not statistically significant. On the other hand, combining MP reduced significantly the number of nodules in the CRC+MP and DCRC+MP groups to 3.33 ± 0.56 and 3.67 ± 0.88 polyp (^#^
*p* < 0.05, * *p* < 0.05), respectively (Figure 6).

It is worth noting that the addition of metformin to probiotics treatment had a better effect in reducing polyp formation, as the total number of polyps in the MP treated groups was significantly lower compared to the P-treated group in nondiabetic CRC animals with ^†^
*p* < 0.05 (Figure 6). 

Diabetic mice that were not subject to colorectal cancer induction (D group), had a higher number of polyps than their nondiabetic counterparts (NC group), with 3.5 ± 0.56 compared to 0.67 ± 0.33, but this difference was not statistically significant (Figure 6).

### 2.6. Histological Alterations of the Colon Due to Metformin and Probiotics Treatment

The histology of the colon in normal control mice (NC group) showed no alterations; there were straight crypts reaching the muscularis mucosa, intact lining epithelium, normal goblet cells, as well as unremarkable changes in inflammatory cells infiltration (Panel 7A-a) with a score of 0.33 ± 0.21.

In contrast, all AOM/DSS-treated animals, nondiabetic and diabetic (CRC and DCRC groups), showed remarkable histological changes in the colon, indicating inflammation and dysplasia. Animals in these groups had the highest histological alterations, with 19.0 ± 1.35 and 20.0 ± 0.58, respectively (Figure 7C). The main changes detected include epithelial ulceration, cryptitis, crypt abscesses, crypt architecture disarray, inflammatory cells infiltration in the mucosa and submucosa, interruption in muscularis mucosa, and goblet cells depletion, as depicted in Panels 7A-e and 7B-m for CRC and DCRC, respectively.

The analysis of the various histological scores indicated a significant effect of the combination therapy in improving the colonic tissue, both in nondiabetic and diabetic groups. 

In nondiabetic animals, single drug administration in group CRC + M (Panel 7A-f) and CRC + P (Panel 7A-g) ameliorated colorectal crypt structure and significantly reduced the histological score to reach 14.0 ± 1.0 and 11.83 ± 0.31, respectively, with ^#^
*p* < 0.05 when compared to untreated CRC (Figure 7C). Treatment with a combination of metformin and probiotics in group CRC + MP (Panel 7A-h) had an effect that was better than either drug alone, and recorded the lowest histological score (8.67 ± 1.17, ^#^
*p* < 0.05) with significantly reduced alterations.

On the other hand, diabetic animals (D group) showed a histological alteration score of 7.50 ± 1.54, significantly higher than nondiabetic animals (NC group) with ^†^
*p* < 0.05 (Figure 7C), despite the fact that they were not subject to CRC induction, thus shedding light on the damage caused by diabetes alone on the colonic tissue, particularly the increase in inflammatory cells infiltrates, as seen in Panel 7B-i.

Moreover, in diabetic CRC animals, the administration of either metformin alone or probiotics alone in the DCRC + M and DCRC + P groups was not able to noticeably reverse the pathological damage (Panel 7B-n and 7B-o), as the respective histological scores were lowered with no statistical significance (ns, Figure 7C). However, when probiotics were combined with metformin, a significant amelioration of the histological alterations was observed in the DCRC + MP group (Panel 7B-p) with * *p* < 0.05 (Figure 7C).

The structural improvement between treatment with metformin alone (CRC + M, DCRC + M) and metformin with probiotics in the CRC + MP and DCRC + MP groups was statistically significant, ^†^
*p* < 0.05 (Figure 7C). One possible explanation could be that the dysregulated microbiota in CRC could prevent metformin from exerting its protective effects. Correction of this dysbiosis with probiotics proved to be crucial in regulating the anti-inflammatory and anticarcinogenic mechanism of action of metformin.

### 2.7. Proliferation Assessment in Colonic Tissue

To assess the effect of metformin and probiotics on cell proliferation, immunostaining by Ki-67 was performed on paraffin-embedded colon tissue. The highest proliferation indices were observed in CRC and DCRC groups, whereby the ki-67 staining was distributed throughout most of the crypt area, towards the lumen (Figure 8).

Metformin administration in nondiabetic and diabetic CRC animals was able to significantly reduce the cellular proliferation in CRC + M and DCRC + M groups (^#^
*p* < 0.05, * *p* < 0.05 respectively). However, the decrease in proliferation index with probiotics administration alone in the CRC + P and DCRC + P groups was not statistically significant (Figure 8C).

When combined, metformin and probiotics (in the CRC + MP and DCRC + MP groups) were the most effective in bringing ki-67 count close to that of the normal controls (^#^
*p* < 0.05, * *p* < 0.05, respectively), indicating that this combination was most likely able to suppress tumor cell proliferation in the AOM/DSS-induced CRC model (Figure 8).

### 2.8. Modulation of Reactive Oxygen Species Production 

To analyze ROS levels within the intestinal epithelium, frozen colon sections were stained with the ROS-responsive dye dihydroethidium (DHE). A significant increase, reaching its highest staining intensity, was obtained in untreated CRC animals in both nondiabetic and diabetic groups (CRC and DCRC) as depicted in Figure 9. Moreover, diabetic animals without CRC induction (group D) showed high ROS production, with ^†^
*p* < 0.05 when compared to the normal controls in the NC group (Figure 9), thus indicating the increased ROS generation in the colon epithelium in diabetes and shedding light on the common oxidative stress increase linking CRC, inflammation, and diabetes. 

In nondiabetics, treatment of cancerous animals with metformin alone, probiotics alone, or their combination significantly reduced ROS production, with the lowest DHE/DAPI ratios obtained in the CRC + MP group, ^#^
*p* < 0.05 (Figure 9).

On the other hand, diabetic animals with CRC showed a significant reduction in their DHE/DAPI ratios only when the combination protocol was adopted in the DCRC+MP group (Figure 9).

### 2.9. Modulation of Nitric Oxide Levels with Probiotics and Metformin

The serum nitric oxide concentration was determined by measuring nitrite concentration, a stable metabolite of NO (Figure 10). Nitrite levels were significantly upregulated in both nondiabetic and diabetic CRC mice. This upregulation was modulated by the different treatments to various extents. In the nondiabetic subgroups, treatment with metformin alone or its combination with probiotics were able to significantly reduce nitrite levels in the CRC + M group and, most importantly, in the CRC + MP group with ^#^
*p* < 0.05, whereby the combination of the two drugs had additive effects in suppressing nitrite production to a level close to that of normal animals (NC group) (Figure 10).

However, in diabetic CRC animals, probiotics administration alone in the DCRC + P group or in combination with metformin in the DCRC + MP group significantly reduced nitrite levels (* *p* < 0.05), without a significant additive effect (Figure 10). On the other hand, diabetic animals (D group), had high levels of nitrite when compared to the normal controls (NC group); however, the difference was not statistically significant. These diabetic animals treated with metformin alone, probiotics alone, or their combination in groups D + M, D + P and D + MP expressed no significant difference and they had comparable nitrite levels among the groups that were higher than normal controls in group NC (Figure 10).

### 2.10. Modulation of IL-6 and TNF-α Production by Metformin and Probiotics 

IL-6 and TNF-α, two critical inflammatory mediators involved in the stimulation of tumor microenvironment, were measured in both serum and colon extracts. 

AOM/DSS treated animals in nondiabetic and diabetic groups (CRC and DCRC) showed elevated IL-6 and TNF-α levels in their colons (Figure 11a,c) and sera (Figure 11b,d). On the other hand, normal controls (NC group) presented significantly lower levels of cytokines when compared to CRC and DCRC, with ^#^
*p* < 0.05 and * *p* < 0.05, respectively.

Treatment with metformin alone or probiotics alone decreased the levels of IL-6 (Figure 11a,b) and TNF-α (Figure 11c,d) in both colonic tissue and serum of cancerous nondiabetic and diabetic animals, as seen in groups CRC + M, CRC + P, DCRC + M and DCRC + P; however, no reduction was observed in the colonic tissues of group DCRC + P. Such reductions were not statically significant.

Importantly the MP combination significantly reduced the IL-6 levels in the colon of nondiabetics (CRC + MP, ^#^
*p* < 0.05) and the serum of diabetics (DCRC+MP, * *p* < 0.05), as seen in Figure 11a,b.

In addition, TNF-α levels were also significantly reduced in both colon and serum of nondiabetics with CRC as seen in Figure 11c,d (CRC + MP, ^#^
*p* < 0.05), and only in the serum of diabetics (DCRC + MP, * *p* < 0.05).

Furthermore, the control diabetic group that did not receive any treatment (group D) also showed high levels of IL-6 and TNF-α production in colon tissue and serum compared to the NC group, shedding light on the inflammatory state created by diabetes in the colon, which will in turn affect the carcinogenic process. 

The synergistic effect between metformin and probiotics was remarkable to various extents in the colon and serum of CRC + MP and DCRC + MP groups. Interestingly, supplementation of MP recovered the healthy levels of these two cytokines, resulting in a significant improvement in the inflammatory response and a consequently lower likelihood of carcinogenesis.

## 3. Discussion

A growing body of evidence sheds light on the association between diabetes mellitus and CRC. The mechanisms underlying these two medical conditions are not fully elucidated yet; however, hyperglycemia coupled with an increase in oxidative stress and chronic inflammation create a favorable environment for the progression of diabetes, IBD, and CRC. Importantly, an altered gut microbiota is being recognized as a key player in this crosstalk [3,32].

In recent years, increasing attention has been given from the scientific community to experimental and clinical studies supporting the role of probiotics in the management of colorectal carcinogenesis and diabetes. Along this line, numerous potential mechanisms of action of probiotics have been proposed, including amelioration of the gastrointestinal mucosa, changes in the intestinal microbiota and in its metabolic activity, modulation of the immune responses, improvement of glycemic parameters, inhibition of cellular proliferation, induction of apoptosis, and exerting anti-inflammatory and antioxidative effects, among others [33,34]. 

This study used the combination of probiotics and metformin in nondiabetic and diabetic Balb/c mice that were subject to CRC induction. Analysis of the emanating data from clinical, histological, and molecular sources pointed to the beneficial effects of the therapy in the various groups, to different extents, in reestablishing homeostatic equilibrium and in actively preventing the inflammatory and carcinogenic processes.

Concerning the protocol, it is worth mentioning that the intended animals successfully developed diabetes based on multiple clinical criteria: high serum glucose >250 mg/dL and excess urination. Moreover, selected groups of these diabetic as well as nondiabetic mice underwent successful CRC induction by AOM/DSS as documented by clinical and histological data. This animal model has been widely used by our group and by other investigators in testing drug efficacy to screen new therapeutic treatments relevant to human IBD and CRC. It is also well established that this model provides reproducible and long-lasting colonic damage that mimics the human colitis-associated CRC process [35] and that DSS-induced chronic inflammation of the colon plays a major role in colorectal carcinogenesis by altering multiple parameters, including the microstructure of the gastrointestinal tract, the intestinal barrier integrity and function, the production of ROS and its metabolites, and the secretion of specific mediators and cytokines. Such changes happen in parallel with a modification in the intestinal microbiota and the creation of a state of dysbiosis [36]. 

On the other hand, diabetes mellitus is one of the most prevalent and rapidly increasing comorbid conditions. For more than 10 years, medical literature has shed light on the relationship between diabetes and CRC, connecting diabetes onset to poor cancer outcomes as comorbid diabetes worsens the course of chronic inflammatory diseases and complicates its management [32,37]. This is in agreement with our results where cancer was exacerbated by diabetes; higher histological score and a worse clinical profile were obtained in nontreated animals that were subject to diabetes induction along with CRC than their nondiabetic counterparts. Moreover, survival rates in diabetic CRC mice (DCRC group) were significantly lower than their nondiabetic counterparts (CRC group).

The administration of a mixture of probiotics along with metformin helped in inhibiting the damage caused by the administration of AOM/DSS to Balb/c mice, through preventing weight loss, ameliorating the DAI, reducing the production of polyps, ameliorating colon histology, and regulating the secretion of the proinflammatory cytokines, thus reducing or inhibiting the inflammatory pathways in the colon. These results are in agreement with numerous previous reports focusing on probiotics and metformin and their beneficial effects on diabetes and CRC [29,31]. These health benefits could be due to the multiple mechanisms involved in such a mutualistic approach of probiotics and the intestinal barrier, resulting in the inhibition of the dynamics of initiation and development of IBD and CRC.

It was proven that dysbiosis in the gut acts as a driving force during the progression from inflammation to carcinogenesis [38,39]; thus, probiotics have the possibility of retaining tumor progression by manipulating the intestinal microbiota and improving multiple related parameters. In this study, metformin or probiotic single drug treatment significantly decreased blood glucose levels reduced glycemia, in comparison to the untreated diabetic mice, with or without CRC induction. These results are in line with preliminary interventions in humans suggesting that probiotics may improve glucose metabolism, insulin, and HbA1c levels [40,41].

In addition, our results suggest a potential chemopreventive effect of probiotics supplementation on CRC, whereby probiotics promoted intestinal homeostasis and regulated the inflammatory response. Animals treated with probiotics alone (CRC + P and DCRC + P) had a better clinical profile when compared to the untreated (CRC and DCRC groups) animals, and their DAI scores were ameliorated with better survival rates. In addition, occult blood appearance was decreased and delayed; polyp formation, inflammatory cells infiltration, and Reactive oxygen and nitrogen species (RONS) secretions were reduced when probiotics were administered. These results parallel those of Mendes et al. where probiotics supplementation reduced inflammatory cell infiltration and lowered the inflammatory response [39].

On the other hand, metformin, one of the most prescribed molecules in the drug market is widely used for the treatment of diabetes mellitus. Metformin is being now recognized as a complex drug possessing antitumor and antiaging effects, as well cardiovascular and neuroprotective properties [37,42,43]. 

The efficacy of metformin in alleviating inflammation and oxidative stress as well as prevention of CRC has been shown to be mediated mainly through the inhibition of various proinflammatory mediators and oxidative stress [44]. These observations were in parallel with our results, whereby metformin administration to CRC animals ameliorated their clinical profile, their DAI scores, and their survival rates. Furthermore, it reduced histological alterations scores and significantly lowered the oxidative stress in CRC + M and DCRC + M animals when compared to the untreated CRC and DCRC groups. 

Collectively, our results showed that the administration of metformin alone or probiotics alone had beneficial effects on the diabetic and CRC phenotypes to variable extents. These variations might be explained by the interindividual differences in the composition of gut microbiota, especially in the inflamed microenvironment created by the induction of CRC and diabetes.

Several recent studies have shed light on the gut microbiota as a key site of action for metformin. This was supported by old data indicating that the efficacy of metformin is affected by antibiotics [45]. Moreover, the glucose-lowering effects were found to be stronger following intraduodenal versus intravenous administration of metformin [46]. 

In the present study, the combination of probiotics with metformin helped metformin in potentiating its anticancerous and anti-inflammatory effects. One possible mechanism might be through the correction of dysbiosis by probiotics, which enhanced the activities of metformin. In fact, animals with CRC and DCRC that were treated with the combination therapy showed a significant amelioration of their diabetic and cancer status when compared to groups treated with a single drug and to untreated groups. 

Cell proliferation is considered as a crucial factor influencing carcinogenesis progression. In our study, immunohistochemical analysis of colon tissue from the different groups showed high levels of Ki-67-labelled cells in the crypts of CRC and DCRC animals; moreover, the ki-67 labelling extended to most of the crypt surface. This is in accordance with other studies showing that in CRC, a reversal in the distribution of proliferating cells from the bottom of the crypt into the upper crypt and luminal surface occurs [47]. However, in normal conditions, proliferating cells are concentrated at the bottom half of the crypt, and the upper half of the crypt usually consists of nondividing migrating cells [48]. Metformin and probiotics administration induced an inhibition of proliferating colonocytes especially when combined, shedding light again on their protective effect on colorectal carcinogenesis.

The beneficial effects granted by metformin and probiotics were also substantiated by the histological analysis of the colonic mucosa. Histopathologic analysis of colorectal tissues showed that the multiple histopathological alterations recorded during the course of the disease were reversed to significant extents, including surface erosion, inflammatory cells infiltration, submucosal edema, polyp formation, and dysplasia. The MP combination attenuated the severity of colorectal inflammation and ameliorated colorectal crypt structure as evaluated in the histological score. It looks like the combination treatment could stabilize the intestinal wall through a mechanism that is different from that of metformin alone or probiotics alone. It is very likely that the integrity of the mucosa needed a different mechanism. Thus, through the balanced microbiota, the junctional complexes of the epithelial cells were maintained; the secretory part of the balanced microbiota could have provided anti-inflammatory elements and protected the mucosal barrier.

Moreover, gut barrier dysregulations and pathologies promote the production of proinflammatory cytokines (IL-6 and TNF-α), which in turn trigger subclinical inflammation and insulin resistance, shedding light again on the inflammation loop between CRC and diabetes [3]. In our study, there was an upregulation of IL-6 and TNF-α in CRC animals with or without diabetes induction in the CRC and DCRC groups. This upregulation was not restricted to colon tissue, as the levels of these cytokines were also upregulated in the serum of the animals, thus emphasizing the systemic inflammation occurring in diabetes and CRC. A strong inhibition of these cytokines was detected in the colon of cancerous animals when probiotics and metformin were combined. It is well established that inflammatory changes in the colonic mucosa are characteristic features of CRC that include infiltration of inflammatory cells and enhanced production of a panel of cytokines [49]. Following recruitment, neutrophils get activated and produce large amounts of proinflammatory mediators, mainly IL-1β and IL-6 [50]. These changes, coupled with elevated levels of RONS generated in the colon tissue, contribute to destructive mucosal damage, which will create a leaking mucosal barrier that could allow multiple bacteria, including toxic strains, to infiltrate and grow, leading to chronic inflammation, an optimal environment for colitis-associated CRC [51].

In the present work, excessive generation of free radicals was indicated by the increased levels of RONS following CRC and diabetes induction in Balb/c mice. These levels were restored to normal by the administration of the combination therapy of metformin and probiotics. The increased nitric oxide (NO) production correlates with the increase in the levels of proinflammatory mediators, such as TNF-α and IL-6, thus leading to exacerbation of the inflammatory chronic reaction at the core of the IBD-CRC etiology [52]. Treatment with metformin in combination with probiotics was able to reduce the release of inflammatory mediators alongside its antioxidant effects, and to reestablish the colonic structure and function of the intestinal wall. 

## 4. Materials and Methods 

### 4.1. Animals

In this study, a total of 96 six-week-old male Balb/c mice were grouped by body weights and were housed in medium sized polysulfone cages at a constant temperature (21 °C ± 2 °C) with an alternating 12 h light/dark cycle. Animal chow (Teklad-Envigo) and water were provided ad libitum. All animal experiments adhered strictly to institutional and international ethical guidelines of the care and use of laboratory animals, and personnel handling animals were qualified. The experimental protocol was approved by the Institutional Animal Care and Use Committee, American University of Beirut, Lebanon (IACUC#16-04-370).

### 4.2. Experimental Design

The animals were randomly divided into two main groups, diabetics and nondiabetics, and two other subgroups, CRC and non-CRC. The animals were then distributed according to the different treatment combinations to form a total of 16 subgroups: Nontreated, treated with metformin (M) alone, probiotics (P) alone, and a combination of the two treatments (MP), as illustrated in Table 2. Mice were individually labelled for tracking, and the average group weight was equilibrated to eliminate any significant weight difference between groups. 

### 4.3. Induction of CRC 

Azoxymethane (AOM)/Dextran sulfate sodium (DSS)-induced colon cancer is a well-established model commonly used in experimental colitis-associated CRC studies. An optimized concentration of the proinflammatory agent DSS (Sigma-Aldrich, Thermo Fisher Scientific, Villebon-sur-Yvette, France) was prepared in autoclaved water and administered to animals in their drinking water. Each DSS cycle consisted of seven days of DSS followed by two weeks normal drinking water. Pilot studies were conducted in order to determine the optimal concentration and needed number of DSS cycles since the colitogenic effect of DSS is affected by several environmental, batches used, and strain-related factors [53]. The carcinogen, AOM (Sigma-Aldrich, Thermo Fisher Scientific, Villebon-sur-Yvette, France), was injected intraperitoneally at the Maximum Tolerated Dose (MTD) of 10 mg/kg body weight.

### 4.4. Induction of Diabetes 

Streptozotocin (STZ) (Sigma-Aldrich, Fisher Thermo Fisher Scientific, Villebon-sur-Yvette, France) at a single dose of 150 mg/kg was used to induce diabetes mellitus. Immediately before administration, STZ was suspended in citrate buffer (pH 4.4–4.5) and injected intraperitoneally [54,55].

### 4.5. Probiotics and Metformin Administration

The probiotic (P) used is a mixture of seven strains of lactic-acid-producing bacteria: *Lactobacillus rhamnosus*, *Saccharomyces boulardii*, *Bifidobacterium breve*, *Bifidobacterium lactis*, *Lactobacillus acidophilus*, *Lactobacillus plantarum*, and *Lactobacillus reuteri*. It was administered with daily dose of 10^8^ CFU per animal. Metformin (Glucophage) treatment was provided at a dosage of 150 mg/kg body weight. Continuous treatments were administered via drinking water since day 1 until the end of the experiment. Their consumption was measured on a daily basis and fresh solutions were administered twice a week. 

### 4.6. Clinical Course Assessment

During the experimental period, body weight, stool consistency, and gross bleeding scores were recorded. A previously validated clinical disease activity index (DAI) ranging from 0 to 4 was calculated based on the following parameters: stool consistency (0, normal; 2, loose; 4, diarrhea), gross bleeding (0, absence; 2, blood stained; 4, presence) and weight loss (0, none; 1, 1–5%; 2, 5–10%; 3, 10–20%; 4, >20%) as per the formula: DAI = (Stool consistency + Fecal bleeding + Weight loss)/3 [56]. 

### 4.7. Blood Glucose Determination

Blood glucose levels (BGL) were measured in tail vein blood using an Accu-Chek^®^ Performa blood glucose meter system. The range for the Accu-Chek is 10–500 mg/dL and any value >500 mg/dL registers as “HI” (Readings of “HI” were recorded as 500 mg/dL). Measurements were done prior to diabetes induction and weekly after STZ injection. Diabetes was diagnosed with BGL >250 mg/dL. All blood glucose measurements were taken in the fed state early in the morning to eliminate variability in blood glucose levels caused by feeding patterns of the mice [57].

### 4.8. Fecal Occult Blood Measurement

Collection of feces was done by placing a single mouse in an empty cage without bedding material for few min; feces were collected and occult blood was measured using Guaiac fecal occult blood test kit, as per the manufacturer instructions [58].

### 4.9. Blood and Serum Collection

On the day of sacrifice, bleeding was performed by cardiocentesis in accordance with approved institutional animal ethical protocols. The blood was collected in BD Microtainer tubes and centrifuged at 2500 rpm for 10 min. The separated serum was stored at −20 °C.

### 4.10. Dissection

At the indicated time point, animals were sacrificed; their colon was isolated, quickly flushed with cold phosphate-buffered saline (PBS) on ice to remove feces and blood. A portion of it was fixed in 10% buffered formalin and the other portion was stored in liquid nitrogen for further analyses. 

### 4.11. Histological Studies

Formalin-fixed descending colons biopsies were paraffin embedded, cut into 5 μm sections on a glass slide, and stained with hematoxylin and eosin (H&E) for general morphology. These protocols were performed in accordance with the standard histology procedures developed by our team [59]. The different sections were photographed using an Olympus CX41 microscope. Histologic scoring was performed on H&E stained colon tissue on a scale adapted and modified from Hussein et al. where seven parameters were evaluated, as listed in Table 3. Each parameter had four scores based on the degree of structural change, accordingly the measures ranged from 0 (normal) to 21 (severe alterations) [60].

### 4.12. Cellular Proliferation by Immunohistochemistry Using Ki-67 Stain

Immunohistochemistry was performed on paraffin-embedded sections. For antigen retrieval, slides were immersed in citrate buffer (pH 6). After washing with TBST, the slides were incubated with a primary antibody (Ki-67, EnCor Biotechnology Inc, Gainesville, FL, USA, 1/1000) at 4 °C overnight. After washing the slides three times with TBST, the sections were incubated with Goat anti-Rabbit IgG Highly Cross-Adsorbed Secondary Antibody, Alexa Fluor 594 (Thermo Fisher Scientific, Villebon-sur-Yvette, France, 1/1000) in TBST with 5% BSA for 2 h at room temperature. The nuclei were counterstained with DAPI and slides were photographed using Zeiss Axio microscope. Sections were evaluated focusing on longitudinally oriented crypts and the number of Ki-67-positive cells per HPF (40× objective) were counted [61,62].

### 4.13. Reactive Oxygen Species Measurement by Dihydroethidium (DHE)

DHE was performed on frozen tissues. Colon rings were demarcated with a solvent-resistant pen. DHE solution (Thermo Fisher Scientific, Villebon-sur-Yvette, France, 1/1000) was dispensed over the tissue. The slides were placed for 30 min at 37 °C and then the DHE residues were removed, slides counterstained with DAPI in a mounting medium, coverslipped, and stored at 4 °C (light sensitive) until microscopic evaluation and quantification using Zeiss Zen 2.3 software (Zeiss, Ulm, Germany).

### 4.14. Determination of Nitrite Levels

Serum nitrite concentrations were measured using the classic colorimetric Griess reaction. One hundred microliter serum samples were pipetted into 96 well microtiter plates, 100 µL Griess reagent (equal volumes of 1% sulphanilamide and 0.1% and naphthylethylenediamine dihydrochloride) was added. After incubation in dark for 10 min at room temperature, absorbance was measured at 550 nm. Nitrite concentration (µM) was calculated from a sodium nitrite standard curve freshly prepared in distilled water [63,64].

### 4.15. Assessment of Cytokine Levels

The levels of IL-6 and TNF-α were measured in both plasma and colon extraction using ELISA assay performed according to the manufacturer’s instructions (Thermo Fisher Scientific, Villebon-sur-Yvette, France). The optical density was measured at a wavelength of 450 nm using a microtiter plate reader (Multiskan Ascent 96/384 plate reader, Thermo Fisher Scientific, Villebon-sur-Yvette, France). The final results were expressed as pg/mL and the limit of detection of IL-6 and TNF-α were 4–500 pg/mL and 8–1000 pg/mL, respectively. 

### 4.16. Statistical Analysis

Statistics were performed using GraphPad Prism 8.0.1, San Diego, CA, USA and data were expressed as a mean ± SEM. Significant differences were evaluated using the one-way ANOVA followed by Tukey–Kramer multiple comparisons test. A value of *p* < 0.05 was considered significant [65].

## 5. Conclusions

Data in this study lead to multiple conclusions that can assist in the development of targeted therapies in the presence of metformin and probiotics. Data suggest that selective cytokine inhibition, as well as ROS and NO inhibition might be an important strategy for the prevention of CRC. Metformin combined with probiotics prevented AOM/DSS-induced damage through attenuating the inflammation pathway in colorectal mucosal cells and reducing tumor cell proliferation, thus leading to the inhibition of colitis-associated CRC. The mechanism supporting these inhibitory effects of metformin might relate to its interaction with the balanced microbiota in the presence of probiotics (Figure 12). However, very little is currently known about the bacterial targets of metformin and it is possible that the microbiota could regulate some of its effects on host physiology via unknown mechanisms. This undoubtedly warrants further investigation. 

## Figures and Tables

**Figure 1 cancers-12-01857-f001:**
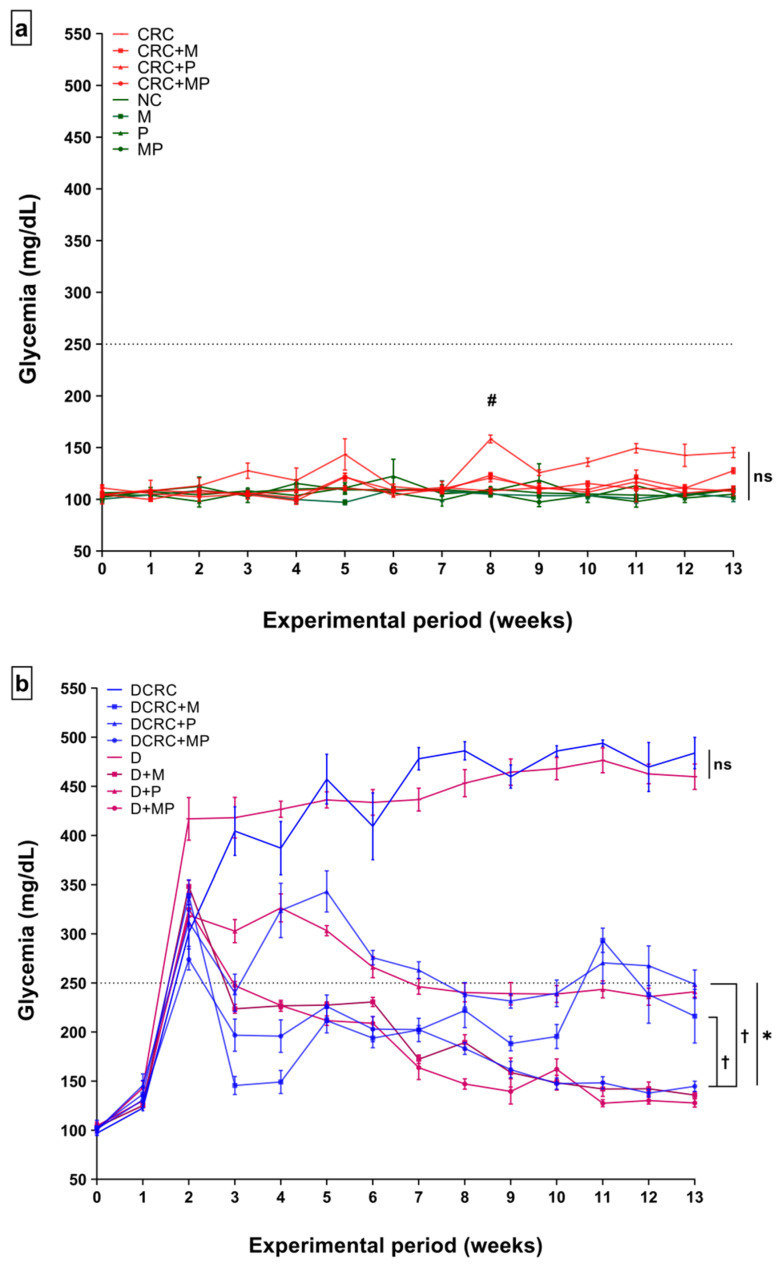
Effects of probiotics and metformin on glycemia levels in nondiabetic (**a**) and diabetic (**b**) Mice. Statistical significance was expressed by ^#^
*p* < 0.05 in nondiabetics when compared to their experimental colorectal cancer control (CRC group) and * *p* < 0.05 in diabetics when compared to their experimental control, the diabetic colorectal cancer mice (DCRC group) with *n* = 6 animals per group. Moreover, when comparing only two groups, connecting lines were used to indicate the compared groups with ^†^
*p* < 0.05, (ns) indicates nonsignificant.

**Figure 2 cancers-12-01857-f002:**
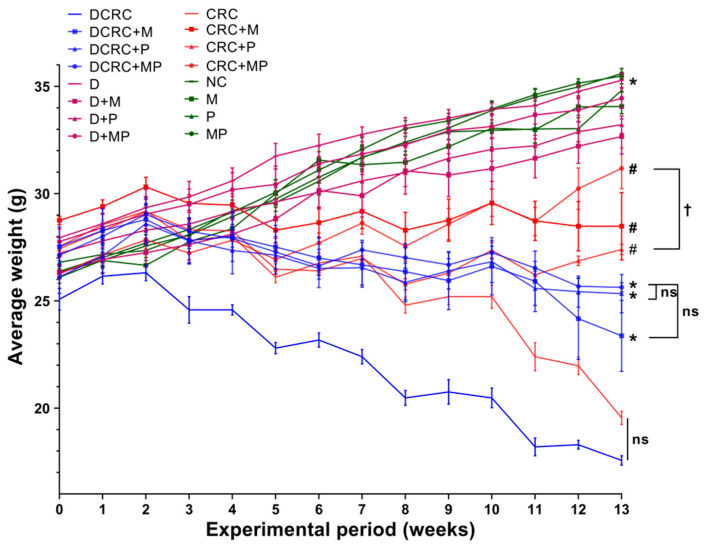
Weight changes in the different animal groups during the experimental period. Data are shown as mean ± SEM with *n* = 6 animals per group. Statistical significance was expressed at week 13 by ^#^
*p* < 0.05 in nondiabetics when compared to their experimental CRC control (CRC) and by * *p* < 0.05 in diabetics when compared to their experimental diabetic CRC control (DCRC). Moreover, when comparing only two groups, connecting lines were used to indicate the compared groups with ^†^
*p* < 0.05.

**Figure 3 cancers-12-01857-f003:**
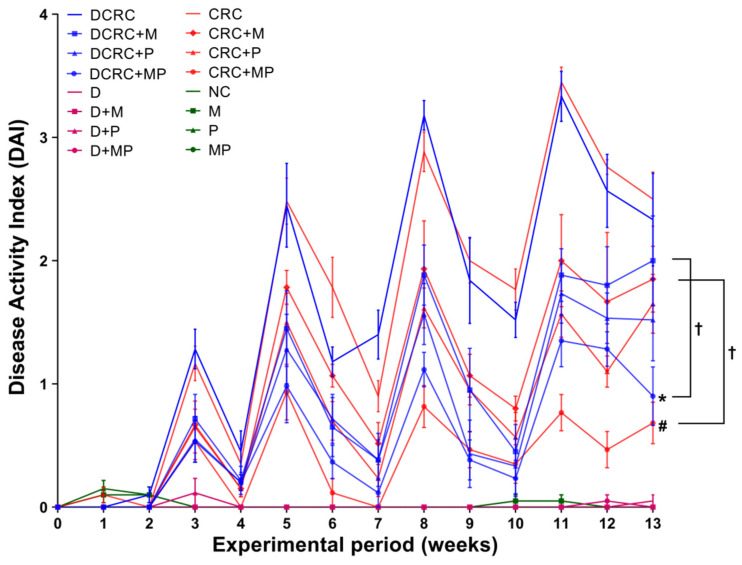
Weekly Disease Activity Index (DAI) variations among the different groups. Note that animals that were not subject to the induction of CRC had very low indexes, close to zero. However, an increase in DAI was obtained in the groups with CRC induction, with peaks obtained at week 3, 5, 8, and 11. M and P treatments elicited a reduction in DAI. Statistical significance was expressed by ^#^
*p* < 0.05 in nondiabetics when compared to their experimental CRC control (CRC), and by * *p* < 0.05 in diabetics when compared to their experimental diabetic CRC control (DCRC). Moreover, when comparing only two groups, connecting lines were used to indicate the compared groups with ^†^
*p* < 0.05.

**Figure 4 cancers-12-01857-f004:**
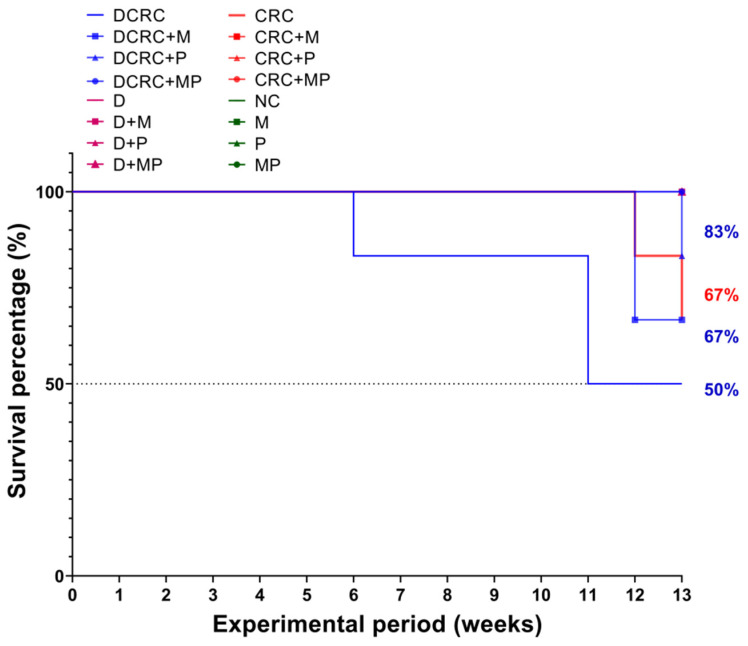
Kaplan–Meier survival curves of animals in the different groups. Diabetic animals with CRC had the worst survival rates of 50% (DCRC group); treatment with M alone or P alone (DCRC+M and DCRC + P groups) raised the survival percentages to 67% and 83%, respectively. In nondiabetic CRC animals, the CRC group had a survival rate of 67%, and no deaths were observed in all other groups.

**Figure 5 cancers-12-01857-f005:**
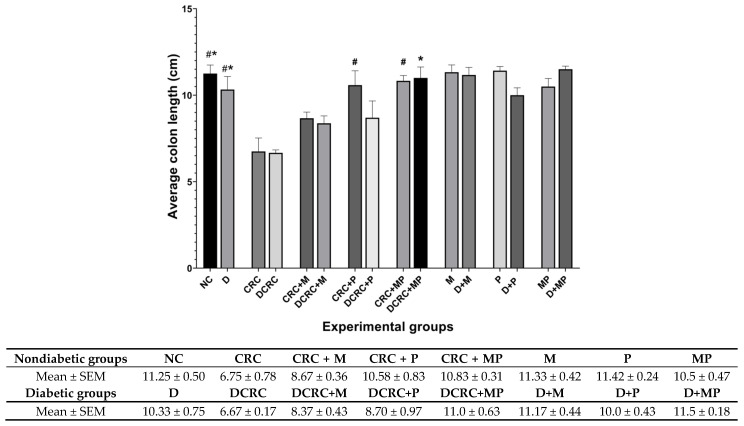
Colon length variation. Average colon length (cm) was recorded on the day of sacrifice at week 13 Nondiabetic groups were compared to their experimental CRC control (CRC), significance was expressed by ^#^
*p* < 0.05. On the other hand, diabetic groups were compared to their experimental diabetic CRC control (DCRC), and significance was defined as * *p* < 0.05, with *n* = 6 animals per group.

**Figure 6 cancers-12-01857-f006:**
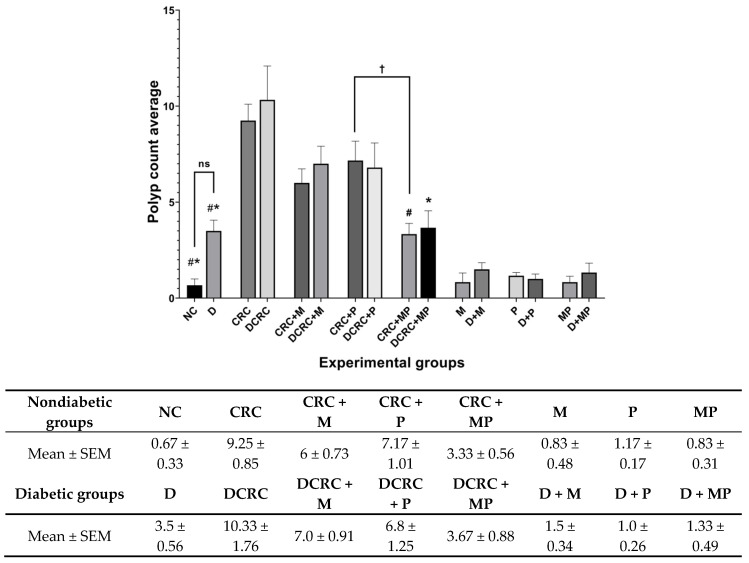
Polyp count variation in the different groups. Average polyp number was recorded on the day of sacrifice at week 13. Nondiabetic groups were compared to their experimental CRC control (CRC). Significance was expressed by ^#^
*p* < 0.05. On the other hand, diabetic groups were compared to their experimental diabetic CRC control (DCRC). Significance was expressed by * *p* < 0.05. Moreover, when comparing only two groups, connecting lines were used to indicate the compared groups with ^†^
*p* < 0.05, (ns) stands for nonsignificant, with *n* = 6 animals per group.

**Figure 7 cancers-12-01857-f007:**
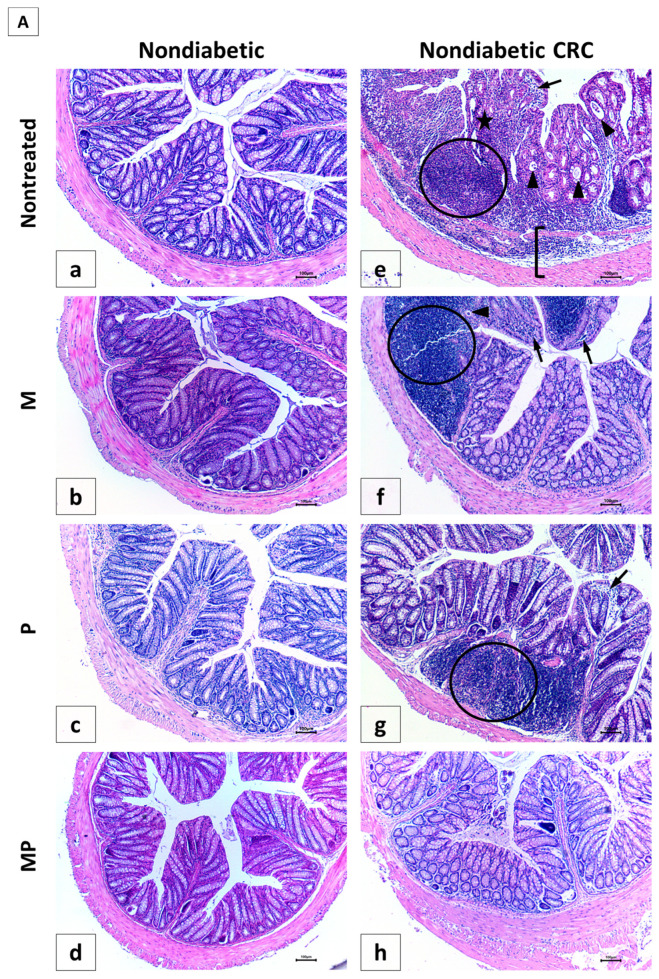
Effect of probiotics and metformin on colon histology. (**A**,**B**) Representative images of hematoxylin and eosin (H&E)-stained colon sections illustrating the histological changes in the nondiabetic (**Panel A**) and diabetic (**Panel B**) groups. The worst alterations were encountered in the nontreated CRC (7A-e) and diabetic CRC (7B-m). Note the presence of large inflammatory cells infiltrates (circle), inflammatory cells invading the edematous submucosa (bracket), crypt abcess (black triangle) and cryptitis (black arrow), as well as crypt architecture disarray (star). Significant improvements in the combination treated mice in groups CRC + MP (7A-h) and DCRC+MP (7B-p) were noted. Original magnification: 4×; scale bars 100 µm. Photos were adjusted for white balance using Adobe Photoshop^®^; (**C**) Histological alterations score. Data is expressed as average ± SEM (*n* = 6). Nondiabetic groups were compared to their experimental CRC control (CRC), significance was expressed by ^#^
*p* < 0.05. On the other hand, diabetic groups were compared to their experimental diabetic CRC control (DCRC), significance was expressed by * *p* < 0.05. Moreover, when comparing only two groups, connecting lines were used to indicate the compared groups with ^†^
*p* < 0.05; (ns) stands for nonsignificant.

**Figure 8 cancers-12-01857-f008:**
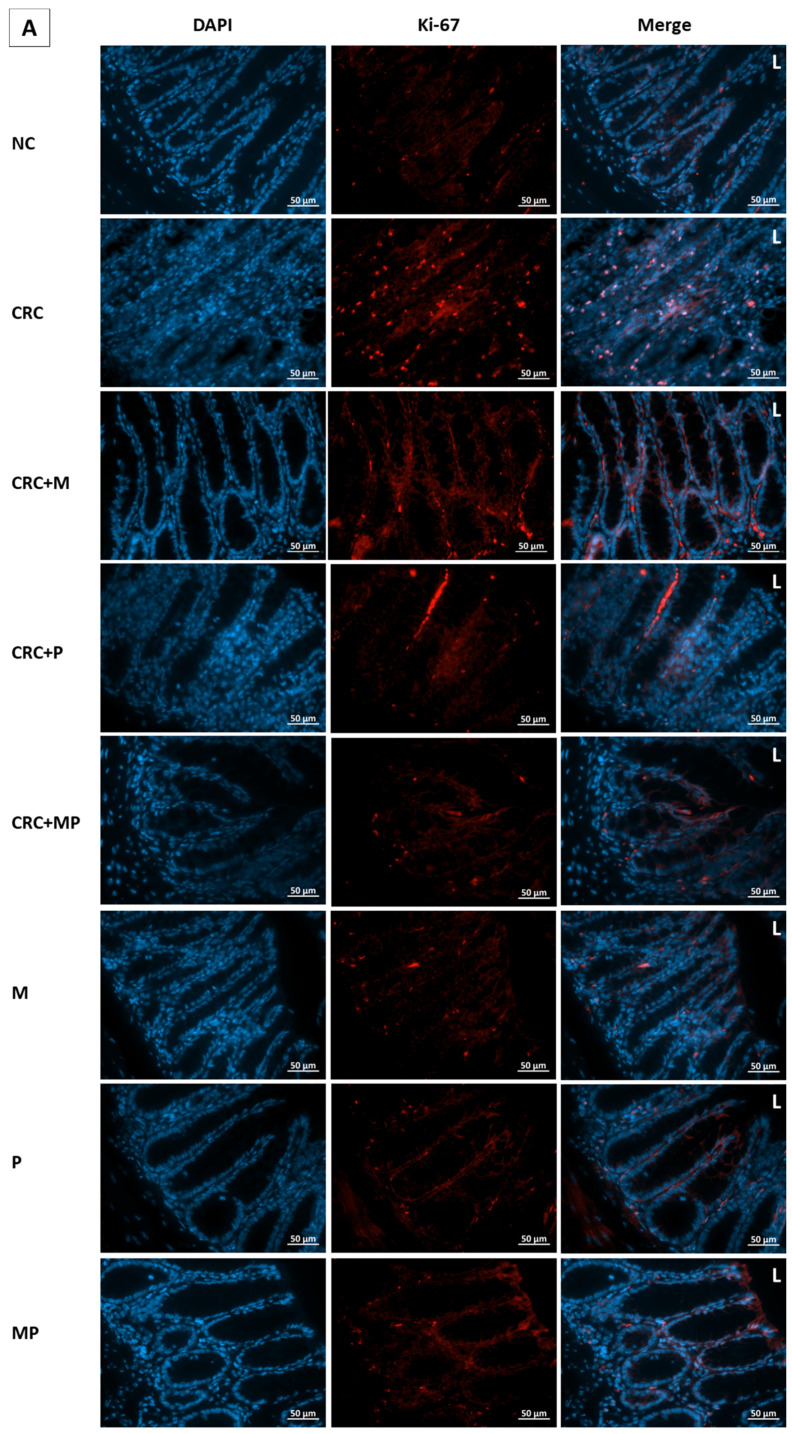
Effect of probiotics and metformin on colon proliferation. (**A**,**B**) Panel of representative images of colon section labelled with with Ki-67 (red) and counterstained with DAPI (blue) in nondiabetic (**A**) and diabetic groups (**B**). HPF 40× magnification; scale bars 50 µm. “L” indicates the position of the lumen. CRC and DCRC groups showed the highest positive nuclear Ki-67 staining throughout the crypts, with increased staining intensity from the midcrypt region to the lumen. A decrease proliferating cells was observed with probiotics and metformin administration to different extents. Normal controls and animals that were not subject to CRC induction had low proliferation indices; (**C**) Proliferation index in the different groups. Data is expressed as mean ± SEM (*n* = 5). Nondiabetic groups were compared to their experimental CRC control (CRC), significance was expressed by ^#^
*p* < 0.05. On the other hand, diabetic groups were compared to their experimental diabetic CRC control (DCRC); significance was expressed by * *p* < 0.05. Moreover, when comparing only two groups, connecting lines were used to indicate the compared groups with ^†^
*p* < 0.05; ns stands for nonsignificant.

**Figure 9 cancers-12-01857-f009:**
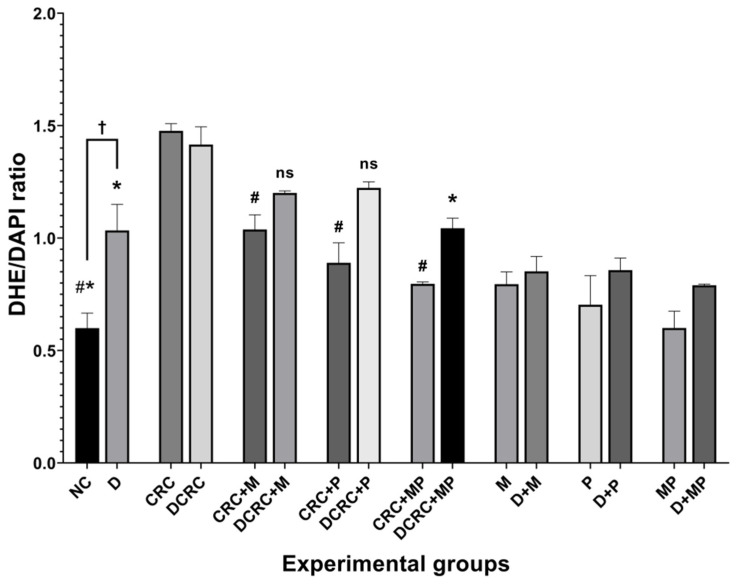
Reactive oxygen species modulation by metformin and probiotics. Results are expressed as ratios of DHE and DAPI intensity (*n* = 3). Nondiabetic groups were compared to their experimental CRC control (CRC group); significance was expressed by ^#^
*p* < 0.05. On the other hand, diabetic groups were compared to their experimental diabetic CRC control (DCRC group); significance was expressed by * *p* < 0.05. Moreover, when comparing only two groups, connecting lines were used to indicate the compared groups with ^†^
*p* < 0.05; ns stands for nonsignificant.

**Figure 10 cancers-12-01857-f010:**
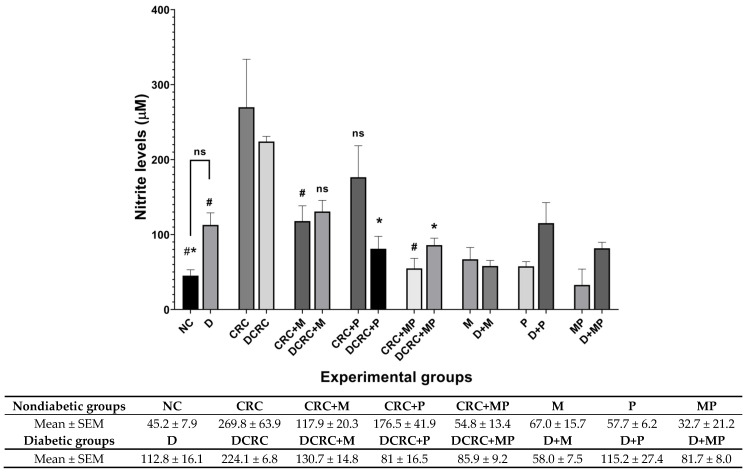
Nitrite modulation by metformin and probiotics. Results are expressed as mean ± SEM (*n* = 4). Nondiabetic groups were compared to their experimental CRC control (CRC), significance was expressed by ^#^
*p* < 0.05. On the other hand, diabetic groups were compared to their experimental diabetic CRC control (DCRC); significance was expressed by * *p* < 0.05; ns stands for nonsignificant.

**Figure 11 cancers-12-01857-f011:**
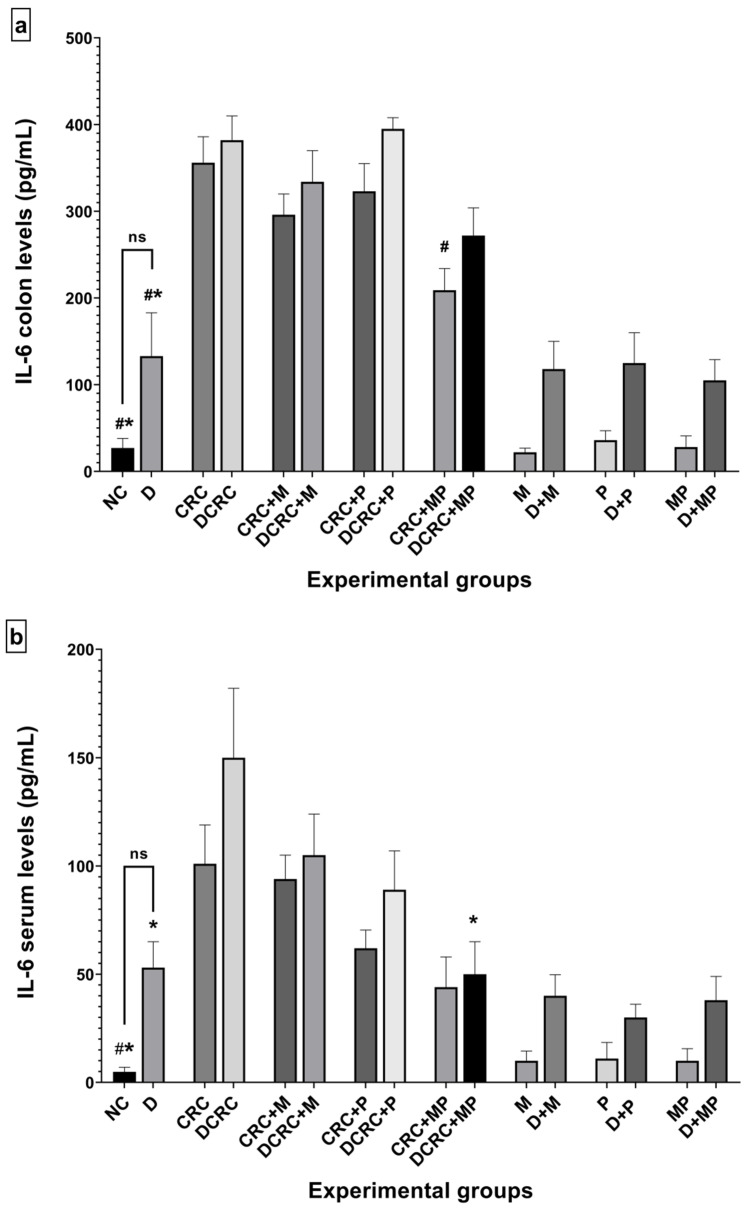
Variation of IL-6 and TNF-α levels from colon extraction (**a**,**c**) and serum (**b**,**d**) of the different groups. Results are expressed as mean ± SEM (*n* = 4). Nondiabetic groups were compared to their experimental CRC control (CRC); significance was expressed by ^#^
*p* < 0.05. On the other hand, diabetic groups were compared to their experimental diabetic CRC control (DCRC group); significance was expressed by * *p* < 0.05; ns stands for nonsignificant.

**Figure 12 cancers-12-01857-f012:**
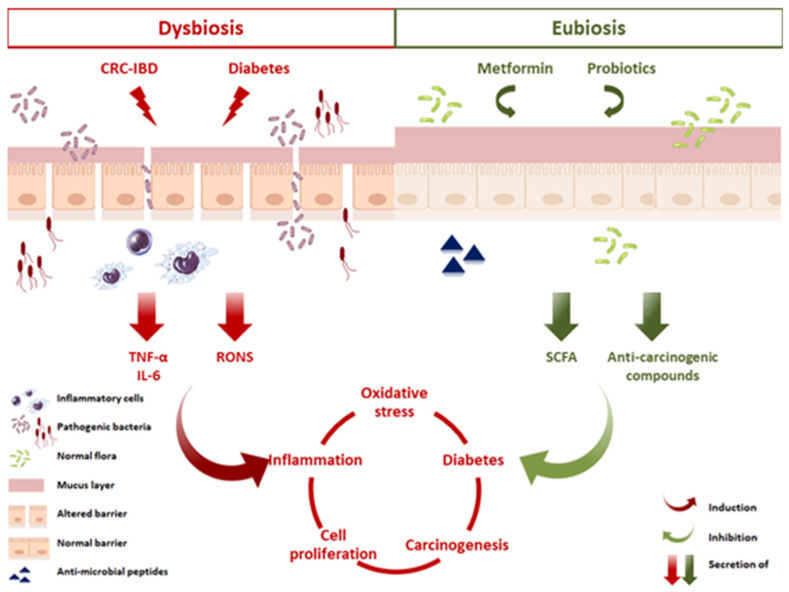
Proposed mechanisms of action elicited by metformin and probiotics in CRC and diabetes. An altered microbiota in a state of dysbiosis impairs the functions of the gut barrier by affecting the tight junctions and the mucus layer, thus facilitating the translocation of pathogens and toxins into the lamina propria. This invasion leads to recruitment of inflammatory cells, their activation, and the secretion of proinflammatory cytokines, including IL-6 and TNF-α. In parallel, an increase in RONS production induces a state of chronic inflammation, DNA damage, and increased cell proliferation, known as key players in CRC progression. Moreover, this state of chronic inflammation promotes insulin resistance and disurbance in glucose homeostasis, thus exacerbating diabetes and enhancing CRC. Probiotics and metformin administration, however, inhibited CRC progression, reduced inflammation, and ameliorated diabetes. These benefical effects are potentially linked to a restoration of the gut barrier, production of SCFA, antimicrobial peptides, regulation of hepatic glucose production, and modulating the balance between proliferation and apoptosis. The result of such a balanced miroenvironment is to preserve a dynamic intestinal barrier that controls and maintains homeostasis.

**Table 1 cancers-12-01857-t001:** Effects of Metformin and Probiotics on Fecal Occult Blood (FOB). FOB was Assessed Weekly; Percentages were Obtained by Calculating the Number of “Positive” Animals in Each Group. A Color Scale Ranging from Strong to Faint Red is Used, whereby the Shade of the Red Color Represents the Value of the Cell.

Group	Experimental Period (Weeks)													
	0	1	2	3	4	5	6	7	8	9	10	11	12	13
NC	0%	0%	0%	0%	0%	0%	0%	0%	0%	0%	0%	0%	0%	0%
D	0%	0%	0%	0%	0%	0%	0%	0%	0%	0%	0%	0%	0%	0%
CRC	0%	0%	0%	17%	33%	100%	100%	50%	100%	100%	100%	100%	100%	100%
DCRC	0%	0%	0%	17%	17%	100%	100%	80%	100%	100%	100%	100%	100%	100%
CRC + M	0%	0%	0%	0%	0%	100%	83%	33%	100%	83%	67%	100%	100%	100%
DCRC + M	0%	0%	0%	0%	0%	100%	33%	17%	100%	50%	33%	100%	100%	100%
CRC + P	0%	0%	0%	0%	0%	83%	33%	33%	100%	100%	67%	100%	100%	100%
DCRC + P	0%	0%	0%	0%	0%	100%	50%	33%	100%	33%	33%	100%	100%	60%
CRC + MP	0%	0%	0%	0%	0%	67%	17%	0%	67%	50%	33%	67%	67%	50%
DCRC + MP	0%	0%	0%	0%	0%	67%	17%	17%	100%	17%	17%	100%	100%	33%
M	0%	0%	0%	0%	0%	0%	0%	0%	0%	0%	0%	0%	0%	0%
D + M	0%	0%	0%	0%	0%	0%	0%	0%	0%	0%	0%	0%	0%	0%
P	0%	0%	0%	0%	0%	0%	0%	0%	0%	0%	0%	0%	0%	0%
D + P	0%	0%	0%	0%	0%	0%	0%	0%	0%	0%	0%	0%	0%	0%
MP	0%	0%	0%	0%	0%	0%	0%	0%	0%	0%	0%	0%	0%	0%
D + MP	0%	0%	0%	0%	0%	0%	0%	0%	0%	0%	0%	0%	0%	0%

**Table 2 cancers-12-01857-t002:** Experimental Design. Balb/c Male Mice were Divided into two Main Groups, Nondiabetics and Diabetics, and Two Other Subgroups, CRC and non-CRC. They were Then Divided According to the Different Treatment Combinations to form a Total of 16 Subgroups.

**Nondiabetic Animals**	
**With CRC Induction**	**Without CRC Induction**
CRC (CRC)	Normal controls (NC)
CRC + metformin (CRC + M)	Metformin (M)
CRC + probiotics (CRC + P)	Probiotics (P)
CRC + metformin and probiotics (CRC + MP)	Metformin and probiotics (MP)
**Diabetic Animals**	
**With CRC Induction**	**Without CRC Induction**
Diabetic CRC (DCRC)	Diabetic (D)
Diabetic CRC + metformin (DCRC + M)	Diabetic + metformin (D + M)
Diabetic CRC + probiotics (DCRC + P)	Diabetic + probiotics (D + P)
Diabetic CRC + metformin and probiotics (DCRC + MP)	Diabetic + metformin and probiotics (D + MP)

**Table 3 cancers-12-01857-t003:** Histological Changes. A Scale Adapted and Modified from Hussein et al. is Used to Calculate the Scores, Where Seven Listed Parameters were Evaluated. Each Parameter has Four Scores from 0 (normal) to 3 (altered) Based on the Degree of Structural Changes [60].

Structural Change	0	1	2	3
Mucosal architecture	Normal	Focal surface destruction	Zonal surface destruction	Diffuse destruction
Glandular crypt architecture	Absent	Mild atrophy	Atrophy + Branching	Atrophy + Branching + Crypt abscess
Loss of goblet cells	Absent	Mild	Moderate	Extensive
Edema	Absent	Mild	Moderate	Extensive
Crypt abscesses	Absent	Focal	Zonal	Extensive
Inflammatory cells infiltration	Absent	Mild (only Mucosa)	Moderate (to muscularis mucosa)	Extensive (to submucosa and musculosa)
Dysplasia	Absent	Focal	Zonal	Diffuse

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
