# Peer review of "Metformin and Probiotics in the Crosstalk between Colitis-Associated Colorectal Cancer and Diabetes in Mice"

_cancers, 2020, doi:10.3390/cancers12071857_

Round 1

Reviewer 1 Report

The of Al Kattar et al reports interesting results about a preventive effects of metformin and probiotics on tumor development in Colitis Associated Colorectal Cancer and Diabetes in an animal model. The study is well designed and the results are convincing and interesting. The main comment concerns the placement of “Material and Methods” section after the results; the position of this section is crucial in every experimental study in order to avoid confounding notion receipt by readers. Therefore, it is mandatory that this Section precedes the Results. Other comments are:

  1. Figure legends need to be shortened and focused to the main result, which each figure illustrates; more details must be correctly placed in the text.
  2. It is unclear which probiotic formulation was administered and the criteria of its choice.
  3. Ki 67 appears to be expressed along the whole intestinal crypt, while currently it is expressed mainly in the lower third in normal tissue; a comment on this feature is required.
  4. Some abbreviations are not preceded by the full text at the first mention.

Author Response

 Reviewer’s response
Reviewer 1:
Comment 0:
The main comment concerns the placement of “Material and Methods” section after the results; the position of this section is crucial in every experimental study in order to avoid confounding notion receipt by readers. Therefore, it is mandatory that this Section precedes the Results.
Answer 0: we have formatted the article as per “Cancers MDPI” requirements as stated in the instructions for authors.
Comment 1: Figure legends need to be shortened and focused to the main result, which each figure illustrates; more details must be correctly placed in the text.
Answer 1: legends shortened as much as possible in Figures 1- 3, 5-11, keeping the necessary
main results.
Comment 2: It is unclear which probiotic formulation was administered and the criteria of its choice.
Answer 2: as stated in the materials and methods section: “The probiotic (P) used is a mixture of 7 strains of lactic acid-producing bacteria: Lactobacillus rhamnosus, Saccharomyces boulardii, Bifidobacterium breve, Bifidobacterium lactis, Lactobacillus acidophilus, Lactobacillus plantarum and Lactobacillus reuteri. This commercially available mixture was used because Lactobacillus and Bifidobacterium are considered, based on published data, as the most commonly administered probiotics with beneficial effects on the gastrointestinal tract [1,2]. Moreover, the product used is “Probiolife” it is commercially available, however, we did not mention the commercial name in order not to promote the product.
1. Ciorba, M.A. A gastroenterologist's guide to probiotics. Clinical gastroenterology and hepatology: the official clinical practice journal of the American Gastroenterological Association 2012, 10,960-968.
2. Molska, M.; Reguła, J. Potential Mechanisms of Probiotics Action in the Prevention and Treatment of Colorectal Cancer.
Nutrients 2019, 11, 2453.
Comment 3: Ki 67 appears to be expressed along the whole intestinal crypt, while currently it is expressed mainly in the lower third in normal tissue; a comment on this feature is required.
Answer 3: all the mucosal epithelial cells of the large intestine arise from stem cells located at the bottom of the intestinal gland. The lower third of the gland constitutes of intestinal stem cells niche. However, some newly generated cells undergo two to three more divisions as they begin their migration up to the luminal surface, where they are shed about 5 days later ; these are most likely
the few ki-67 positive cells that were expressed in some normal tissues
[3].
3. Ross, M.H. Histology: a text and atlas:with correlated cell and molecular biology, 6th ed.; Lippincott Williams & Wilkins: Phiadelphia, 2011.
Comment 4: Some abbreviations are not preceded by the full text at the first mention.
Answer 4: The following were adjusted in the text
ï‚· probiotics (P) and metformin(M) line 120
ï‚· Streptozotocin (STZ) line 124
ï‚· Dextran sulfate sodium (DSS) line 130
ï‚· metformin and probiotics (MP) line 132

Reviewer 2 Report

In this manuscript entitled “Metformin and Probiotics in The Crosstalk Between Colitis Associated Colorectal Cancer and Diabetes in Mice”, the authors identified that combination of metformin and probiotics could improve diabetes and colitis- associated colorectal cancer in mice by inhibiting inflammatory responses and oxidative stress. Overall, this is an interesting study. The manuscript is well-prepared and the experiments are planned in a systematic way. Only few concerns need to be addressed.

  1. Figures 1-3: The authors should consider to use only four colors to paint the four groups (control, D, CRC, and DCRC), and four marks to label the four treatments (NC, M, P, and MP), which may be easier to interpret the results.
  2. Figures 5, 6 and 9: It would be more informative for readers if the authors provide the representative gross macroscopic images of the colon in each group in these two figures.
  3. Figure 2 (CRC and DCRC), Figure 6, 10, 11 (DC and D): The authors should check the statistical analyses between the indicated treatments in these figures because they seemed to be different from each other. In Figure 6, for example, D was significantly different from NC according to the provided values of Mean, SEM, and N.
  4. Figure 8C is missing.

Author Response

 Reviewer’s response
Reviewer 2:
Comment 1: Figures 1-3: The authors should consider to use only four colors to paint the
four groups (control, D, CRC, and DCRC), and four marks to label the four treatments (NC,
M, P, and MP), which may be easier to interpret the results.
Answer 1: figures adjusted in the manuscript as per your request.
We used 4 different colors: red for CRC groups, blue for DCRC groups, pink for the diabetics and
green for the normal mice.
Additionally, four different marks were used for each treatment, No symbol for the untreated,
square symbol for the metformin treatment, triangle symbol for the probiotics treatment, and a
circle for the combined MP treatment.
Comment 2: Figures 5, 6 and 9: It would be more informative for readers if the authors
provide the representative gross macroscopic images of the colon in each group in these two
figures.
Answer 2:
-Figures 5-6
:
Kindly find below, a panel of gross macroscopic images of the colon form the different groups (see
Figures 1-2). They depict the colon shortening in colorectal carcinogenesis and the improvement
in the colon length with the treatments and most importantly the MP combination.
Such results are presented in the manuscript in the form of a bar graph to facilitate for the reader,
the comparison of the averages and statistical analysis. The authors considered a bar graph as
sufficient with the limited space for the article and size of the gross specimens to include.
Figure 1: Gross macroscopic images of the colon from non-diabetic groups
Figure 2: Gross macroscopic images of the colon from diabetic groups
- Figure 9: Adding the panel of DHE stain doesn’t have an added value since the histological
architecture of the frozen sections as you know, is not very well preserved with lots of histological
artifacts. For sake of comparing the intensity of DHE in the various groups, the bar graph was
adopted.
You will find below, two panels of the representative photos from each group, one panel for
diabetics and another for the non-diabetics for your consideration (see Figures 3-4).
In such panels, the beneficial effect of the treatments on ROS production is shown, as depicted for
instance when examining DHE intensity.
The highest levels were obtained in CRC and DCRC groups, decreasing to various extents with
the treatments and most importantly with the combination in groups CRC+MP and DCRC+MP,
reaching DHE levels close to that of the normal controls.

Figure 3: Representative photomicrographs of DHE stained (red) frozen colon sections, counterstained with DAPI (blue) in non
diabetic animal groups

Figure 4: Representative photomicrographs of DHE stained (red) frozen colon sections, counterstained with DAPI (blue) in
diabetic animal groups

Comment 3: Figure 2 (CRC and DCRC), Figure 6, 10, 11 (DC and D): The authors should
check the statistical analyses between the indicated treatments in these figures because they
seemed to be different from each other. In Figure 6, for example, D was significantly different
from NC according to the provided values of Mean, SEM, and N.
Answer 3: As stated in the materials and methods “Statistics were performed using GraphPad
Prism 8.0.1 and data were expressed as mean ± SEM. Significant differences were evaluated using
the one-way ANOVA followed by Tukey-Kramer multiple comparisons test. A value of P<0.05 was
considered significant”
We have rechecked the statistical analysis by prism software, kindly find below the analysis and
p-value for each figure:
ï‚· Figure 2: In this figure, the statistical significance was assessed at the terminal time
point (week13) - “at week 13” statement is added on line 183.
ï‚· Figure 6: Polyp count variation in the different groups p-value = 0.1734
ï‚· Figure 10: Nitrite modulation by metformin and probiotics. p-value = 0.8142
ï‚· Figure 11: Variation of IL-6 and TNF-α levels from colon extraction (a,c) and
serum (
b,d)
ï‚· TNF –serum : p-value =0.2640
ï‚· TNF –colon : p-value =0.3263
ï‚· IL-6 serum : p-value =0.4138
ï‚· IL-6 colon: p-value=0.3393
Comment 4: Figure 8C is missing.
Answer 4: This is an uploading mistake, figure added to the manuscript.
